# Preliminary Investigation on Systems for the Preventive Diagnosis of Faults on Agricultural Operating Machines

**DOI:** 10.3390/s21041547

**Published:** 2021-02-23

**Authors:** Massimo Cecchini, Francesca Piccioni, Serena Ferri, Gianluca Coltrinari, Leonardo Bianchini, Andrea Colantoni

**Affiliations:** Department of Agriculture and Forest Sciences, University of Tuscia, Via S. Camillo De Lellis, 01100 Viterbo, Italy; cecchini@unitus.it (M.C.); francesca.piccioni@unitus.it (F.P.); serenaferri@unitus.it (S.F.); gcoltrinari@unitus.it (G.C.); l.bianchini@unitus.it (L.B.)

**Keywords:** damage detection, failure, vibration, ultrasonic, fatigue, crack

## Abstract

This paper aims to investigate failures induced by vibrations on machines, focusing on agricultural ones. The research on literature has brought to light a considerable amount of data on the driven vehicles and not much on the operating machines, including the ones that we looked for. For this reason, it was decided to direct a survey with the people who work with agricultural machinery every day: operators, sub-contractors, and producers. They were asked about the most frequent breakage, particularly in relation to the rotary harrow, the topic of this work. The questionnaire results showed the types of failures the harrow is most vulnerable to, indicating the times of failure and reparation and the need to set up a potentially useful preventive maintenance supporting system on these machines. Part of the work was then focused on the proposition of a method to investigate bearing failures in the rotary harrow, considering that these have been analyzed in the technical literature and in the survey as the most at-risk components. The proposed method in this work serves as a beginning for the development of a future on board sent-shore-based maintenance system for continuous monitoring of the bearing.

## 1. Introduction

Rolling bearings, in the industrial application, are considered critical components [1]. In fact, a defect in a bearing, unless detected in time, can cause considerable industrial damage [2] as well as risks to human health. In particular, the phenomenon of vibrations can be potentially dangerous for human health, leading to strong negative effects [3,4]. Approximately more than half of all machine failures are due to bearing faults [5]. Bearing defects can occur during machine use, called distributed defects (surface roughness, waviness, and off-size rolling elements) [6], or during the manufacturing process, called local defects (cracks, pits, and spalls) [7]. The presence of defects leads to an increase in normal emission levels of noise and vibration in the bearings [3,8] and an increase in the level of Whole-Body Vibration (WBV) exposure for the driver of the vehicle [9]. Relevant exposure levels of whole-body can be dangerous for the driver [10] and, at the same time, are annoying during work activity [11]. In order to comprehend the ways of reducing the exposure to the WBV for the driver, many studies have been conducted [12], while other studies have been carried out in order to identify the effects of vibrations on machine components. On the basis of this characteristic, many researchers have studied the problem of rolling bearing defects in different ways. During the tests, some researchers ran bearings up to their failure in order to monitor changes of vibration response [13], some of the others intentionally introduced a defect in the bearing, under test, to evaluate the noise and vibration response, varying the size of the defect [14,15]. Many mechanical strains can occur with components of the machinery generated by many different types of vibration, for example, the torsional vibration of the crankshaft. These fatigues generate problems that are often detected after many work hours and cause crankshaft failure and undesirable engine noise [16]. Monitoring the roller bearing emissions, as an indication of wear, allows operators to manage preventive maintenance of the machine components. This avoids the unplanned maintenance, which takes place only at the breakdowns and which causes much higher repair costs than simply repairing the faulty component if it is detected early [17]. Different methods are used for the detection and diagnosis of bearing defects; they may be broadly classified as vibration and acoustic measurements, temperature measurements, and wear debris analysis [18,19,20]. A lot of studies have been carried out in the literature about the failure of driving machines, also in the agricultural fields [5,21,22]. An advanced system was developed in order to detect and control the frame vibration caused by a high-power engine and to prevent/reduces the potential damage to agricultural machinery [23,24]. It is based on intelligent sensor nodes at the measuring points of the two-stage vibration isolation system with the aim to assess the acceleration values and the frequency spectrums [24,25]. But few works have been conducted with respect to those with the problem of WBV exposure [26]. For this reason, this study takes into account the possible damages that can occur in the rolling bearings of agricultural operating machines, with a particular focus on the rotary harrow. Firstly, a questionnaire was given to workers, who day have to work with agricultural machinery every day, in order to collect information about the failures that usually affect harrows. Then, in order to evaluate the optimum level of operating conditions for the bearing in agricultural machines, we monitored the emitted vibrations through a tri-axial accelerometer fixed on the harrow structure. Beyond this optimum level, preventive maintenance is required. This paper shows the first raw values.

## 2. Materials and Methods

The research started with the formulation of a questionnaire aimed at agricultural operators and manufacturers of agricultural machinery. The purpose of the questionnaire was to determine the points of greatest risk of breakage in the harrow with rigid teeth and whether, effectively, a system of preventive maintenance can be useful (“activity one”). In parallel with the main subject of this study, an experiment was started at the experimental farm “N. Lupori” of the University of Tuscia (Italy), where it was possible to plan the data acquisition activities on rigid-tooth rotary harrows (“activity two”).

### 2.1. Investigation of Faults in Agricultural Machinery: The Rotary Harrow

A questionnaire was developed using Google forms, directed to operators, subcontractors, and manufacturers. The questions were:Can you indicate below the role that best represents you at the present time?What is the average frequency of failures in the power harrow?What is the most common type of fault?In the case of bearing failures, which are the most frequently affected?In the event of a failure, how much do you think the average down-time (time from failure to repair) is?In the most common cases of failure on a rotary harrow, how much do you think the repair costs will be? (Indicate an approximate cost or range of costs, from a minimum of... to a maximum of...).On a scale of 1 (not at all useful) to 5 (fully useful), how important would it be for you to have on board a system that warns the driver in advance of a possible breakdown of the machine (detecting the first signs of deterioration of particular elements well in advance before actual breakdowns occur)?

### 2.2. Description of Investigation Method for Bearing Failures in the Rotary Harrow

In order to characterize the mechanical structure of the harrow and highlight the influence of a vibration source, an experimental campaign was carried out in the same boundary conditions. The aim of this activity was to identify the influence of the signal source, represented by the tractor, on the vibration levels of each bearing. Due to the long time needed for signal acquisition and data processing for each test cycle, the measurements were repeated for several days. The experimental samples were acquired on three different tractors, Fiat 55l, New Holland TLK75FA 2005, and Lamborghini 135, as shown in Figure 1.

The measurements were obtained on two-wheeled tractors, and one tracked. The technical characteristics of each vehicle are shown in Table 1.

The measurement of vibratory levels of the harrow bearings was defined with the aim of identifying a methodology that is as repeatable as possible. The harrow under investigation was characterized by the presence of six bearings, distributed in a symmetrical way with respect to the Power Take-Off (PTO) with three bearings per side. The numbering has been defined by observing from behind the machine, identifying nr. 1 as the first bearing on the left and in progression up to nr. 6, last on the right, as shown in Figure 2.

The seventh bearing waqs assigned to the element mounted above the PTO. The positioning of the instrument is shown in Figure 3.

Measurements were realized for each bearing from number 1 to number 6; then the last acquisition was performed on the seventh bearing while keeping the angular speed of PTO constant with the following values: 600 rpm, 1000 rpm, 1500 rpm, and 2000 rpm. The measurement was carried out in the following way:Start-up of the instrument to measure the signal for the bearing being evaluated;Interruption of the measurement by the “pause” command;The shift of the accelerometer to the next bearing;Restart the measurement.

The acquisition time for each bearing was 60 s, except for the Lamborghini 135 with 120 s of measurement time. The data acquisition started when the engine speed was constant in order to avoid recording the transient signal between engine idle and the constant speed defined at the specific level, as described above. Then, seven intervals of time were extracted from the total measurement recorded. Every interval corresponded to each harrow bearing, and each data was classified according to the numbers from 1 to 7. This measurement technique was the best to obtain, from a single acquisition, both an overall assessment of the phenomenon of all mechanical elements of the harrow and the specific characterization of each bearing. The measurements were repeated three times in sequence for each level at a constant speed, with a break of about 20 min between tests, in order to avoid the excessive overheating of the surface. The instrument employed in this experimental campaign was a triaxial 393 PCB (PCB Piezotronics Inc., Depew, NY, USA) accelerometer (0.1 mV/ms^−2^). This device was specially designed for vibration analysis in vehicle testing, instrumentation, and equipment monitoring. It was mounted on the rigid metallic cover of the bearing and, before the positioning, the surface was cleaned to ensure better adherence of the instrument. The accelerometer was connected to a Sound-book MK2 system (SINUS Messtechnik GmbH, Leipzig, Germany), and the data was processed with the Noise and Vibration Work software (Spectra S.r.l., Vimercate, Italy). Subsequent to the recording of measurements, the data was saved in the sound-book device and finally was stored on a hard disk for the analysis. The sampling frequency was set to 1250 Hz. Once the setup of the sound-book was ready, a preliminary recording with minimal engine rpm was done in order to establish the efficiency and stability of the entire system, especially the engine of the tractor. For each signal, there has been obtained a graph of the Time History (trend of the signal to the variance of the time) and the frequency spectrum (one-third-octave bands), both for the overall measurement for each of the single bearings. Moreover, the values were also reported in terms of the equivalent level Leq, both in time as well as in energy content at various frequencies. Ultimately, the analysis was executed as an overall evaluation of the measurement to evaluate the stability of the source signal and for each individual bearing to define the specific strain. For instance, some illustrative graphs are presented in “Results”. The study was structured for analysis of the signal trend over time (Time History) to assess the bearing of any anomalous components due to diverse constituents, such as the instability of the hydraulic system that supports the harrow, which has generated sudden impulsive components. The measures were seen according to the procedure described above on the tractors in different phases due to the availability of vehicles on the farm.

## 3. Results

### 3.1. Questionnaire Preliminary Investigation

Ninety-four percent of the interviewees answered the questionnaire. The results showed that the failures along the rotary harrow occur at least one time per year. The favorable setting for the failure was represented by the rolling bearings or the tooth, close to the knives near the gear case. When a failure occurred, the average time needed for the repair was around 2–3 days, and the cost was about 900 euros. Figure 4 indicates the level of agreement of the interviewees for the implementation of an on-board system for preventive maintenance on a rotary harrow: the y-axis shows the percentage of agreement between the interviewees, and the x-axis is the level of utility to be 1 (low interest) to 5 (high interest).

### 3.2. Experimental Investigation

The experimental activity, conducted on the power harrow below investigation, was based on the analysis of the amplitude values and the frequency spectrum of the vibratory signal with the aim of characterizing the strains of each rolling bearing. The standardized measurement methodology was set with different steps: each measurement was repeated three times for each rolling bearing; it was carried out at different engine rpm, and the data acquired was analyzed at once, after the measurements, in order to infer the accuracy and the rightness of the process. The graphs in the following Figure 5, Figure 6, Figure 7 and Figure 8 show the acquisition data of the experimental measurement, comparing the signal (Time History) and frequency spectrum for each tractor. The blue line indicates the signal, vibration from the Fiat 55l, the red line from New Holland TLK75FA 2005, and the green line of the Lamborghini 135 tractors.

Table 2 shows the maximum values for each bearing in terms of velocity (mm/s) and frequency (Hz) for each direction (x, y, and z). The data is compared for each angular speed of the engines, which was defined before the trial. These values are more representative of the typical daily tractor activity.

## 4. Discussion

With the intent of demonstrating the applicative usefulness of a preventive maintenance system applied to agricultural operating machines, a questionnaire was conducted, considering the rotary harrow as a reference machine. This questionnaire, aimed at agricultural workers and machine makers, first highlighted the most frequent types of breakage, then highlighted the need to have a fault detection system on board. Moreover, the investigation has shown that the greatest number of failures concern the rolling bearings inside the machine’s kinematic chain. The measurement method proposed to assess the level of damage through emitted vibration of rolling bearings seemed useful. Acceleration along three axes in the direction of each bearing mounted on the harrow was measured with three different vibration sources represented by the tractors, as reported above. The measurement technique was repeated for each configuration with constant rpm at different stages. The purpose of the experimental campaign was to compare the frequency spectrum and vibration levels of each bearing in order to evaluate the consequence of any abnormal components due to wear. Results showed a reduction of acceleration values at the increasing of rpm, especially for the rolling bearings settled on the right side of the rotary harrow under investigation. Furthermore, the data obtained showed a different level of bearing wear due to the characteristics of the mechanical transmission from the PTO to each sprocket wheel. For each bearing, the signal recorded with the bearing showed a higher r.m.s. (root mean square) value compared to previous measurements at 1000, 1500, and 2000 rpm constant engine levels. The graphs of the frequency spectrum described above highlight the characteristics of the mechanical solicitations during the tests. The x-axis reports the values in terms of the third-octave bands. The range, until 2 Hz, was defined by the signal components that can be generated by thermic errors. The temperature of the harrow surface, where the sensor was fixed, could increase after a few minutes. It is possible to observe that the range starting from 63 Hz was defined by the solicitations caused by the rotary movement of spheres inside bearings. Furthermore, the components due to the rotation of the crankshaft and oscillation of the engine mechanical components were between 8 Hz and 31.5 Hz. The Lamborghini 135 tractor was the most stable source with an almost constant signal, while the Fiat 55l model showed a fairly variable signal. In conclusion, the results demonstrate that the tractor represents a source of vibration which strongly influences the measured signal. The Lamborghini 135 tractor was the most stable source with an almost constant signal, while the Fiat model showed a fairly variable signal. This last fact did not allow for careful characterization of the vibration signal.

## 5. Conclusions

An experiment was conducted in order to read the reaction of the bearings to different types of vibration sources. The result indicated the effects of engine speed and the influence of the type of source, represented by different tractors. The data was significantly important for the mean range values of the left and the right accelerometer in the x, y, and z directions. The results highlight that the conditions of the tractor (mass and degree wear) influence the signal vibration: an older tractor engenders a more unstable signal compared to a more recent one. It is necessary to point out that, despite the many factors influencing signal stability, measurements have shown acceptable results and encourage further studies regarding these problems. Another experiment could certainly contribute to the creation of a fairly complete database with the intention of accurately determining the characteristics of the vibrations on the bearings. This experiment could be performed in two different ways. Particularly, the tests could be conducted using different tractor models and, at the same time, repeating the measurements on the same machinery described in this work, after a defined period, in order to evaluate the presence of damages on the bearings.

## Figures and Tables

**Figure 1 sensors-21-01547-f001:**
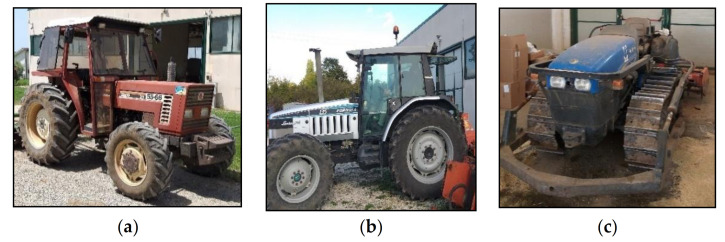
Fiat 55l (**a**), Lamborghini 135 (**b**), and New Holland TLK75FA 2005 (**c**).

**Figure 2 sensors-21-01547-f002:**
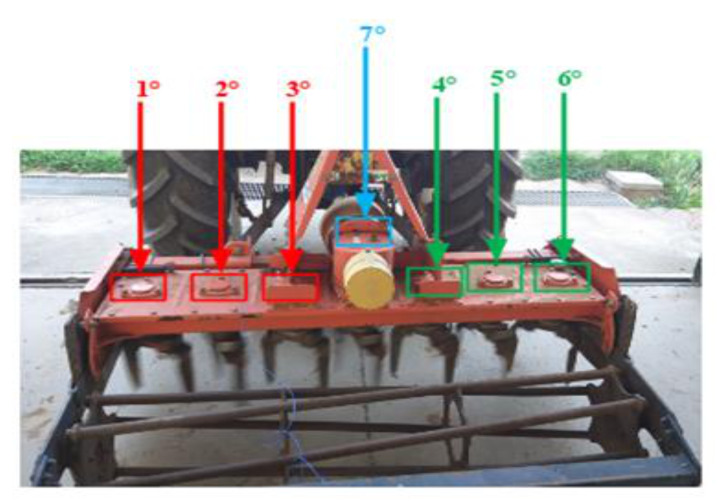
Rotary harrow under investigation.

**Figure 3 sensors-21-01547-f003:**
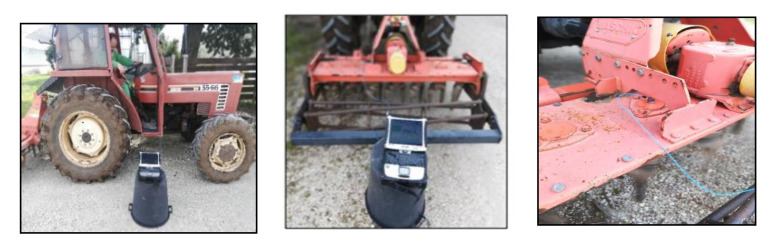
Positioning of the instrument near the machine and accelerometer on the harrow.

**Figure 4 sensors-21-01547-f004:**
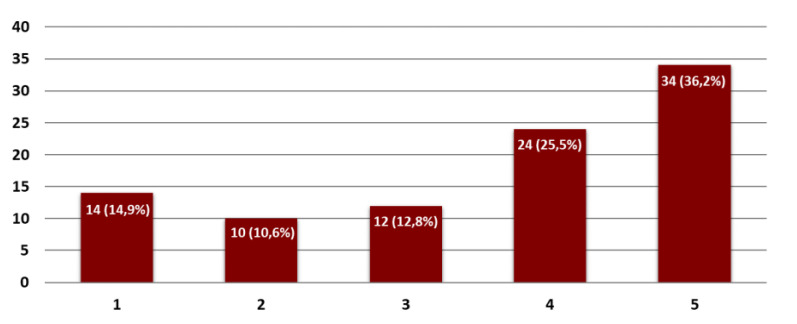
Level of interest for the implementation of an on board system for preventive maintenance on a rotary harrow taken out from the questionnaire.

**Figure 5 sensors-21-01547-f005:**
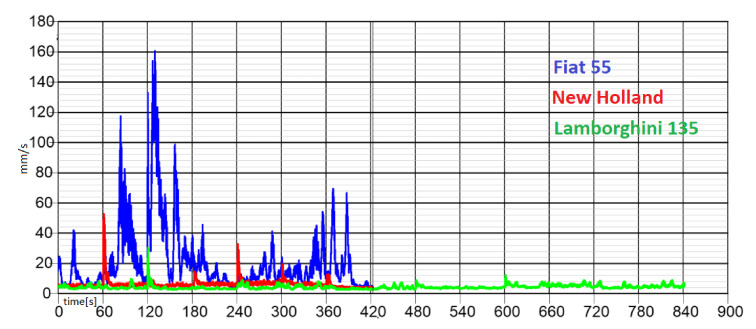
Comparison of time history for all tractors at 1000 rpm.

**Figure 6 sensors-21-01547-f006:**
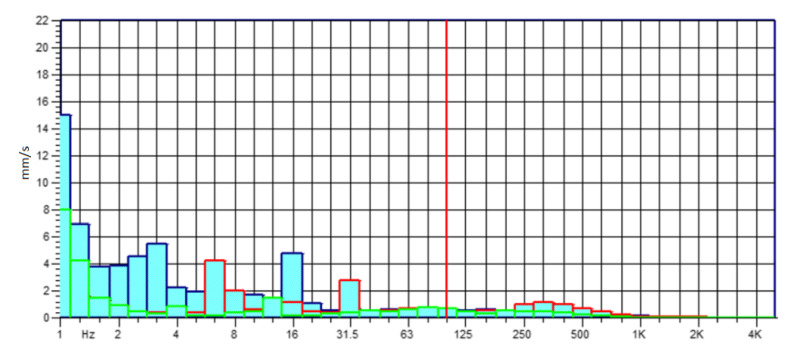
Frequency spectrum for all tractors at 600 rpm—Fiat 55L.

**Figure 7 sensors-21-01547-f007:**
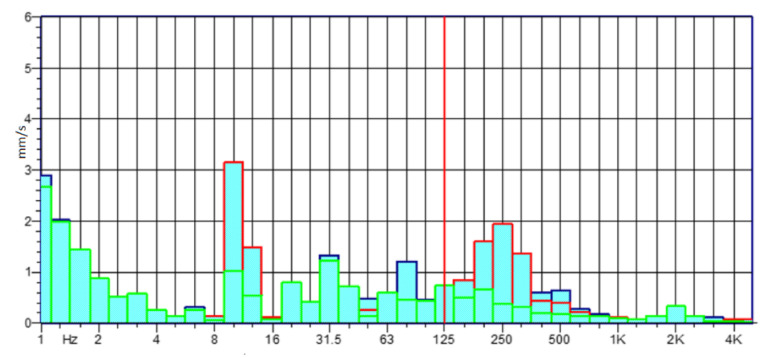
Frequency spectrum for bearing nr. 7 at 2000 rpm—Lamborghini 135.

**Figure 8 sensors-21-01547-f008:**
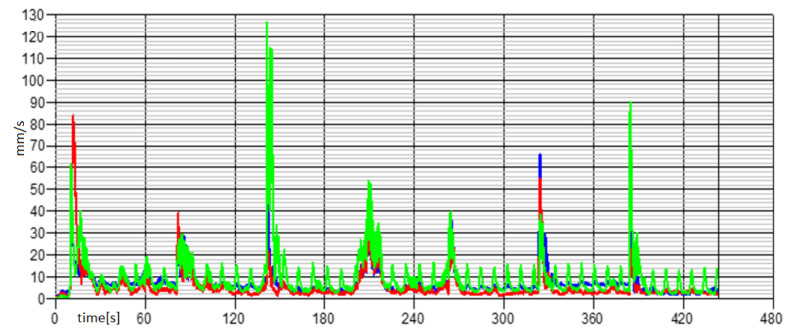
Overall time history at 600 rpm—Fiat 55l.

**Table 1 sensors-21-01547-t001:** Technical characteristic of the tractors.

Tractor	Mass (kg)	Motor Torque (Nm)	Engine Power (kW)
Fiat 55l	2925	176.0	41
New Holland TLK75FA 2005	4150	322.0	63
Lamborghini 135	6100	597.1	100

**Table 2 sensors-21-01547-t002:** Data obtained for the bearings of each vehicle at different engine rpm.

Tractor	Engine Speed	Bearing						
		1°	2°	3°	4°	5°	6°	7°
Fiat 55l	600 rpm	8.55	11.52	24.09	15.92	10.61	9.95	13.02
Fiat 55l	600 rpm	16	16	16	16	16	16	16
Fiat 55l	1000 rpm	10.78	23.44	40.48	10.24	10.17	12.74	14.08
Fiat 55l	1000 rpm	25	16	12.5	12.5	12.5	12.5	12.5
Fiat 55l	1500 rpm	3.38	3.62	3.55	2.49	2.83	2.89	2.04
Fiat 55l	1500 rpm	6.3	25	25	315	250	6.3	160
Fiat 55l	2000 rpm	9.7	5.86	5.40	3.31	3.57	3.88	4.43
Fiat 55l	2000 rpm	31.5	31.5	31.5	31.5	31.5	31.5	31.5
New Holland TLK75FA 2005	600 rpm	6.11	5.02	4.48	4.46	5.96	7.27	4.89
New Holland TLK75FA 2005	600 rpm	6.3	6.3	6.3	31.5	6.3	6.3	6.3
New Holland TLK75FA 2005	1000 rpm	4.40	5.035	3.173	4.30	4.687	4.816	3.74
New Holland TLK75FA 2005	1000 rpm	2.5	2.5	2.5	2.5	2.5	2.5	2.5
New Holland TLK75FA 2005	1500 rpm	5.59	5.48	4.19	4.65	5.29	5.44	5.46
New Holland TLK75FA 2005	1500 rpm	25	25	63	25	25	25	25
New Holland TLK75FA 2005	2000 rpm	6.00	4.98	8.63	9.67	9.98	8.85	10.12
New Holland TLK75FA 2005	2000 rpm	31.5	31.5	31.5	31.5	31.5	31.5	31.5
Lamborghini 135	600 rpm	3.45	2.93	6.12	2.79	8.47	3.32	4.45
Lamborghini 135	600 rpm	12.5	12.5	80	100	125	12.5	4
Lamborghini 135	1000 rpm	3.12	3.11	3.46	2.90	3.25	4.03	3.21
Lamborghini 135	1000 rpm	10	100	80	100	12.5	12.5	250
Lamborghini 135	1500 rpm	415.98	5.30	8.27	3.53	3.79	5.06	3.84
Lamborghini 135	1500 rpm	25	25	25	100	125	315	250
Lamborghini 135	2000 rpm	5.68	6.58	5.87	4.20	5.34	5.10	5.81
Lamborghini 135	2000 rpm	10	10	10	10	125	10	10

## Data Availability

Publicly available datasets were analyzed in this study. This data can be found here: https://drive.google.com/drive/folders/1FKhNAsyfXMIac6_T--mleEEASyDOG6Lv?usp=sharing (accessed on 20 February 2021).

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
