# Peer review of "Preliminary Investigation on Systems for the Preventive Diagnosis of Faults on Agricultural Operating Machines"

_sensors, 2021, doi:10.3390/s21041547_

Round 1

Reviewer 1 Report

General comments:

The topic of this paper is interesting and actual but some revisions are necessary. 

The methodology and the results are clearly presented.

The results obtained are highlighted. English language could be improved.

Introduction should be improved. The literature review should be enriched.

The quality of the Figures and the presentation of the paper should be revised.  

The reference style should be improved.

Minor comments:

ABSTRACT: 

-Line 10 “re-search”..please correct, thank you.

-Line 11 “machines”..please delete the repetition of this word.

-Line 19 why “re-ported”?, please correct.

-INTRODUCTION:

Please add other References with recent publication years to underline the actuality of the research topic. The literature review could be enriched.

Please try to contextualize the problem in the introduction considering the general aim of the topic and the benefit with the obtained results. (Sustainable Development Goal, etc…)

-Lines 24-28 . Please recheck the use of commas in these lines.

-Line 33 “whole-body vibration exposure”  Whole-Body Vibration (WBV)..please correct.

-Lines 35-37. Please improve English in these lines.

-Lines 82. Could be interesting also to have other features of the 3 engines used. Please can you add some information (also with a simple table) of the 3 engines? For example power, efficiency, displacement or emissions?

-Line 83 “Figure 1 and not figure1”.

-Line 105 and 2000 rpm.please put the space.

-Line 121. “WBV”. Please the abbreviation is just defined in Line 33.

-Line 155 “.Seven ..please put the space.

-Line 125. Please can you indicate the tolerance of the instrument used?

-Figure 4 Please improve the quality of Figure and define the x-axis and y-axis in the Figure.

-Line 174 “in the following figure”..please define the number of the Figure.

-Figure 5 .Please put a legend in the Figure in which is indicated the kind of engine for line blu and for line red.

-Why in Figure 6 “Frequency spectrum” and in Figure 7 “frequency spectrum”? please correct.

-Figure 8 . Please improve the quality of the Figure :

Y axis: the unit measurement is too near the number of y axis.

X axis. The name “tempo is in Italian and not in English. Please check.

-Line 189 “the following table 1” . Following it is necessary? Table and not table 1.

REFERENCES:

Please check the style of references, required from the Journal template. All the references have been modified:

  1. Author 1, A.B.; Author 2, C.D. Title of the article. Abbreviated Journal NameYearVolume, page range.
  2. Author 1, A.; Author 2, B. Title of the chapter. In Book Title, 2nd ed.; Editor 1, A., Editor 2, B., Eds.; Publisher: Publisher Location, Country, 2007; Volume 3, pp. 154–196.
  3. Author 1, A.; Author 2, B. Book Title, 3rd ed.; Publisher: Publisher Location, Country, 2008; pp. 154–196.
  4. Author 1, A.B.; Author 2, C. Title of Unpublished Work. Abbreviated Journal Namestage of publication (under review; accepted; in press).
  5. Author 1, A.B. (University, City, State, Country); Author 2, C. (Institute, City, State, Country). Personal communication, 2012.
  6. Author 1, A.B.; Author 2, C.D.; Author 3, E.F. Title of Presentation. In Title of the Collected Work (if available), Proceedings of the Name of the Conference, Location of Conference, Country, Date of Conference; Editor 1, Editor 2, Eds. (if available); Publisher: City, Country, Year (if available); Abstract Number (optional), Pagination (optional).
  7. Author 1, A.B. Title of Thesis. Level of Thesis, Degree-Granting University, Location of University, Date of Completion.
  8. Title of Site. Available online: URL (accessed on Day Month Year).

Author Response

Dear Reviewer 2,

Thank you very much for your interesting observations. 

We reported the following corrections:

  1. the abstract is correct at Lines 10,11 and 19;
  2. the introduction is modified with many new references of the article;
  3. rechecked the use of commas in these lines (24-28);
  4. Whole-Body Vibration correct (33);
  5. improved English (35-37);
  6. a table reports the technical data;
  7. correct the name of Figure;
  8. the tolerance of the instrument is indicated;
  9. the legend is in Figure 5;
  10. correct Frequency Spectrum;
  11. the quality of the figure is improved;
  12. the words "tempo" and following table are correct;
  13. the style of references was checked

Moreover, the English editing of the whole text was checked by a professional service.

Reviewer 2 Report

The authors present an interesting preliminary work (communication article) about the damages suffered by agricultural machinery; however, I consider that the authors have to take into consideration the following suggestions in order to improve your research:

  1. I suggest you follow the journal format.
  2. The quality of figures must be increased.
  3. The figures must be labeled; for example, what is the means of vertical and horizontal axes (fig 4 to 8) ?.
  4. I recommend you increase the description of your frequency graphs, in order to mention the reasons for these frequencies.
  5. I suggest you separate the section results and discussion in two sections (the first one with results and the second one with a discussion) in order to condense all obtained results and mention the importance/advantages of your work in the second section.
  6. I recommend you mention in the conclusion section, what is the next with this investigation? 

Author Response

Dear Reviewer,
Thank you very much for your interesting observations. 

We reported the following corrections:

  1. the paper is corrected according to the format template;
  2. the figures in the paper are modified;
  3. the measurement units are defined on each axis;
  4. the characteristics of the frequency spectrums are described (lines 244-254);
  5. results and discussion are subdivided into two sections;
  6. the conclusions are modified in accordance with your suggestions

Moreover, the English editing of the whole text was checked by a professional service.

Round 2

Reviewer 1 Report

The authors modified the paper according to the reviewers' comments.